# Effects of electroporation on *Acanthamoeba Polyphaga*

**Palloma Santiago Prates Pessoa, Raphael Barcelos, Larissa Fagundes Pinto, Denise de Freitas, Mauro Campos***

Department of Ophthalmology and Visual Sciences, Paulista School of Medicine, São Paulo Hospital, Federal University of São Paulo, São Paulo, São Paulo, Brazil - UNIFESP

* mscampos@uol.com.br

## Abstract

### Background

Species of the genus *Acanthamoeba* spp. are ubiquitous and can cause *Acanthamoeba* keratitis (AK), a serious corneal infection. Due to the toxicity and ineffectiveness of currently available prolonged therapies, we investigated electroceutical treatment aimed at facilitating the permeation of molecules through the membrane of cysts and trophozoites, which allows for faster elimination of the parasite.

### Methods

Cysts and trophozoites of *Acanthamoeba polyphaga* (ATCC® 30461™) were exposed *in vitro* to an electric field with intensities of 2,000 volts and 2,500 volts. Viability after electroporation was assessed by the exclusion method with 0.4% trypan blue dye, while permeabilization was assessed by fluorescence microscopy using propidium iodide (PI), since both are impermeable to the membrane of viable and intact cells. The images were acquired on a Nikon Eclipse TI-U microscope and analyzed using *ImageJ* software.

### Results

With regard to viability, 40% of the trophozoites electroporated at 2,000 V and 42% of those electroporated at 2,500 V were lost, while for cysts the loss was 13% and 16% respectively. Considering permeabilization, 55% of trophozoites and cysts were permeabilized at 2,000 V ($p \leq 0.05$); and 59% at 2,500 V for both ($p \leq 0.05$). Values of $p < 0.05$ were considered statistically significant.

### Conclusion

The voltages tested were effective for both cysts and trophozoites, since the percentages of permeabilization were close, with no statistical significance between them, only with the control groups. These results suggest the possibility that an electroceutical treatment could be applied as a complement to the standard treatment for AK.

**Data availability statement:** All relevant data are within the manuscript.

**Funding:** The author(s) received no specific funding for this work.

**Competing interests:** The authors have declared that no competing interests exist.

## Introduction

Studies show that the genus *Acanthamoeba* has amphizootic characteristics, i.e., they are free-living unicellular protozoa capable of living in various environments, such as water, soil, dust, air, hospital equipment, public and private places, where they occasionally invade hosts, causing disease [1–3].

According to the biological cycle, *Acanthamoeba* has two life cycle stages: the trophozoite and the cyst. The trophozoites are metabolically active and infective, where they multiply, feed and invade host cells. Under extreme conditions of temperature, osmolarity, salinity, pH and lack of food, the trophozoites transform into cysts. These are dormant and resistant forms, enabling them to persist and spread in the environment, and to cause chronic infections in cases of eye disease [1,3].

This protozoan can be classified according to the morphology of its cysts (Pussard and Pons classification, 1977) into three distinct groups including 25 species or based on 18S rDNA genetic sequencing, currently comprising 23 distinct genotypes (T1-T23), [4–9]. Despite their almost universal presence throughout the environment, infections caused by *Acanthamoeba* are relatively uncommon in humans. But the combination of disruption of the corneal epithelium barrier, either through trauma or the use of contact lenses, and exposure to a sufficient inoculum of *Acanthamoeba* substantially increases the risk of keratitis (inflammation of the cornea). As a result, the patient may present with some initial non-specific symptoms, such as tearing, photophobia, blurred vision and intolerance to the use of contact lenses. There may also be blepharospasm, eyelid edema and corneal infiltrate, often confused with other ocular infections, progressing to severe cases with more specific symptoms, including severe painful corneal ulcers that can extend to the inner layers of the eyeball [5,10–20,23,24].

The treatment of *Acanthamoeba* keratitis (AK) is prolonged, intensive and expensive, as it is a resistant etiologic agent. In order to eliminate trophozoites in the initial phase, as well as eradicating cysts and trophozoites during the treatment process, the approach during treatment involves the combination of two distinct classes of drugs: aromatic diamidines, which act on DNA synthesis (propamidine isethionate 0. 1% or hexamidine 0.1%), and cationic antiseptics that induce membrane lysis (biguanide 0.02% to 0.06% or chlorhexidine 0.02% to 0.2%). The most common combination is biguanide 0.02%, with or without propamidine 0.1%. Propamidine is only effective in the trophozoite phase, has no cysticidal properties and is only used in the first month of treatment. Biguanide, on the other hand, is often maintained for long periods of time, with the average duration of treatment varying from four to six months (it can extend from one to 26 months), showing certain efficacy against trophozoites and cysts. However, evidence comparing the effectiveness and safety of these treatments remains limited, with no significant difference detected between chlorhexidine and polyhexamethylene biguanide (PHMB) in clinical outcomes [25]. Sometimes, this type of therapeutic intervention is not able to eliminate the amoebas from the cornea and surgical interventions are necessary, thus demonstrating the urgency of searching for new methods or improvements in treatment, with the aim of reducing the treatment duration for patients [5,13,14,19,21,22,26]. This led to the proposal of possibly incorporating electroceutical treatment in conjunction with the drugs already available.

The electroporation procedure consists of application of electric fields that temporarily alter the permeability and conductivity of the plasma membrane of biological cells. This process takes place inside the electroporator, where the intensity of the electric field, the number of short electrical pulses and the force applied can be adjusted to form hydrophilic pores, allowing the permeation of molecules that normally cannot cross the barrier imposed by the

lipid bilayer. When only used to alternate the entry and exit of molecules, it is a reversible process, i.e., the membrane remains permeabilized for a few minutes and then closes, allowing cellular equilibrium to be re-established. With higher pulse parameters, electroporation becomes irreversible, preventing the cell from restructuring [27–31].

The electroporation technique has been studied as an alternative in the treatment of certain types of cancer, offering clinical efficacy and aesthetic improvements. This approach demonstrates a satisfactory level of safety in its therapeutic use, suggesting that it could be applied in the field of ophthalmology to improve treatments for eye conditions [32–34].

In recent years, these microorganisms have gained the attention of the scientific community due to the increase in the number of cases of eye infections. In this context, this study aims to analyze the opening of pores in the parasite wall in order to possibly facilitate the permeation of drugs through it in the future, which could become an effective treatment for cases of AK.

## Methods

### Description of the isolates of *Acanthamoeba polyphaga*

A standardized *Acanthamoeba* strain of *Acanthamoeba polyphaga* (ATCC® 30461™) was used in this study, obtained from a corneal scrape of a keratitis case in Texas, 1973.

### *Acanthamoeba Polyphaga* culture

The ATCC 30461 isolate was grown on non-nutrient agar medium supplemented with heat-inactivated *E. coli* (DH5α) at 25 °C. After these strains were harvested, the non-nutrient agar plate was washed with transport saline and centrifuged at 2000 RPM for 15 minutes. After centrifugation, the supernatant was discarded and the amoeba pellet resuspended in 1 mL of PYG (proteose-peptone - yeast extract - glucose) culture medium, which was then inoculated into a 25 cm$^2$ culture flask containing 4 mL of culture medium and incubated in an oven at 25 °C. To obtain the trophozoite forms, the isolate was grown in PYG (proteose-peptone - yeast extract - glucose) culture medium. For cystic forms, it was grown in Neff's encystment medium.

### Electroporation

Trophozoites and cysts of the ATCC 30461 isolate were counted in a Neubauer chamber and their concentration was adjusted to obtain an initial inoculum of $5x10^6$ trophozoites/cysts per mL of electroporation buffer (285 mM sucrose, 0.7 mM MgCl$_2$, 1 mM KCl, 10 mM HEPES, 3 mM NaOH). Next, 100 μL of this inoculum (a total of $5x10^5$ amoebas) were added to the electroporation cuvettes, which were then exposed to an electric field of 2000 and 2500 V in a single pulse of approximately 1.4 seconds. Subsequently, fluorescence-based methods for detecting electroporation were carried out as described below in the "Electroporation detection" section, to standardize and evaluate the optimal voltage for *Acanthamoeba* cysts and trophozoites.

### Electrochemical treatment

The *in vitro* experimental model was divided into the following groups:

- Controls (cysts and trophozoites without electroporation.

- Group 1: cysts and trophozoites electroporated at 2000 volts.

- Group 2: cysts and trophozoites electroporated at 2500 volts.

### Electroporation detection

To investigate the viability and permeabilization of electroporated cysts and trophozoites of isolate ATCC 30461, two tests were applied: feasibility and permeabilization, as described below.

### Feasibility analysis

To assess the loss of cysts and trophozoites due to the applied shock, a separate analysis was performed. Amoebas in each group were counted before and after electroporation using 0.4% Trypan blue in a Neubauer hemocytometer. With the numbers obtained from the counts compared to the control groups, it was possible to determine separately the loss by electroporation from the loss of sample by the experimental process.

### Permeabilization analysis

Another analysis was performed using fluorescence intensity. Electroporated cysts and trophozoites were plated in 96-well plates and stained with propidium iodide (PI; Sigma-Aldrich, USA).

The PI dye does not penetrate the wall/membrane of intact cysts and trophozoites, so if the amoebas turn red, it means that there has been a change in the permeability of these structures, allowing the dye to enter, and the greater the uptake of this fluorescence, consequently, the greater the damage to the wall or membrane of this isolate. 4',6-diamidino-2-phenylindole (DAPI; Sigma-Aldrich, USA), acts as a marker, staining the nucleus of any cell blue. In summary, cysts and trophozoites electroporated in electroporation buffer + PI at a ratio of 1:100 (10 μg/mL) were placed in 1.5 ml microtubes in sterile PBS (1X) for 15 minutes at room temperature protected from light and then fixed with 4% paraformaldehyde (PFA; Synth, Brazil) for 15 minutes at 4 °C protected from light, after which they were stained with DAPI dye at a ratio of 1:1000 for 10 minutes at room temperature. The 96-well plates were scanned for fluorescence intensity using an excitation wavelength of 488/617 for PI and 340/488 for DAPI and analysis was carried out using the IN Cell Analyzer 2200 device and counting using *Image J* software. The greater the intensity of the fluorescence captured, the greater the permeabilization. The images were acquired on a fluorescence microscope (Nikon Eclipse TI-U, Nikon, Tokyo, Japan) using the Nikon NIS-Elements platform, in which cysts and trophozoites stained in red (propidium iodide) were considered permeabilized. To reduce variation, all wells were run on the same plate at the same time, and the same batch of PI and DAPI was used for all experiments.

### Statical analysis

Statistical analyses were carried out using the GraphPad Prism 8.0. A two-way ANOVA test was used for multiple comparisons between groups. Values of $p < 0.05$ were considered statistically significant.

## Results

### Feasibility

From the initial inoculum of 200000 amoebas in 100 μL, there is a loss of 106083.3 in the controls due to centrifugation processes. Considering the remaining amount after accounting for losses unrelated to electroporation-induced death, the calculations were made. The results obtained show that 40% of the trophozoites electroporated at 2000 V and 42% of those

electroporated at 2500 V were lost, while for cysts the loss was 13% for 2000 V and 16% for 2500 V, as demonstrated in Fig 1.

## Permeabilization

As for the permeabilization analysis, the results obtained from fluorescence analysis show that permeabilization at 2000 V is 55% for trophozoites and 55% for cysts ($p \leq 0.05$); while at 2500 V it is 59% for trophozoites and 59% for cysts ($p \leq 0.05$), as demonstrated in Fig 2, not being statistically significant between the electroporated groups, only when compared to the control group. Figs 3 and 4 show respectively the permeabilization of trophozoites and cysts.

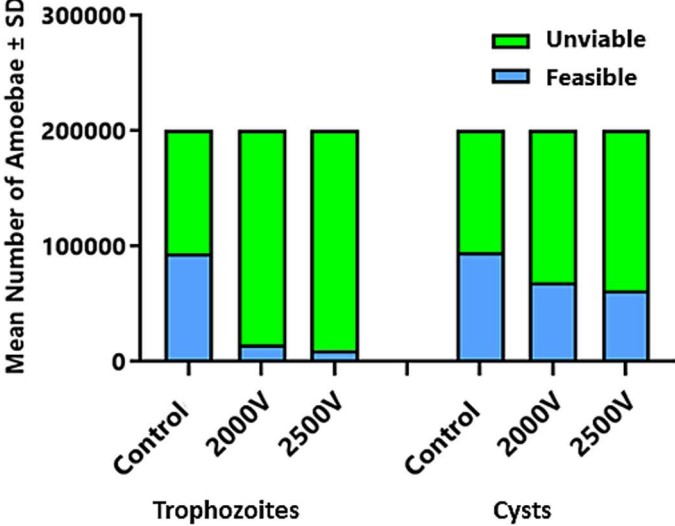

**Fig 1. Demonstration of viable and non-viable cysts and trophozoites in absolute numbers from the Neubauer chamber count.**

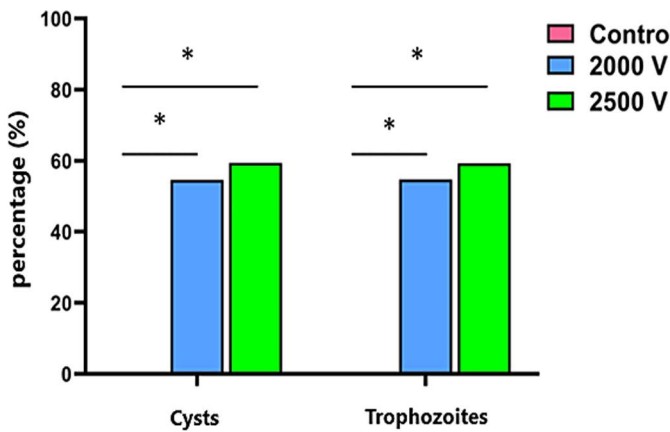

**Fig 2. Permeabilization of cysts and trophozoites of *A. polyphaga* (ATCC 30461) electroporated at 2000 and 2500 volts.** * indicates a significant difference between the control group and the groups electroporated at 2000 and 2500 volts ($p < 0.05$) (Two-way ANOVA).

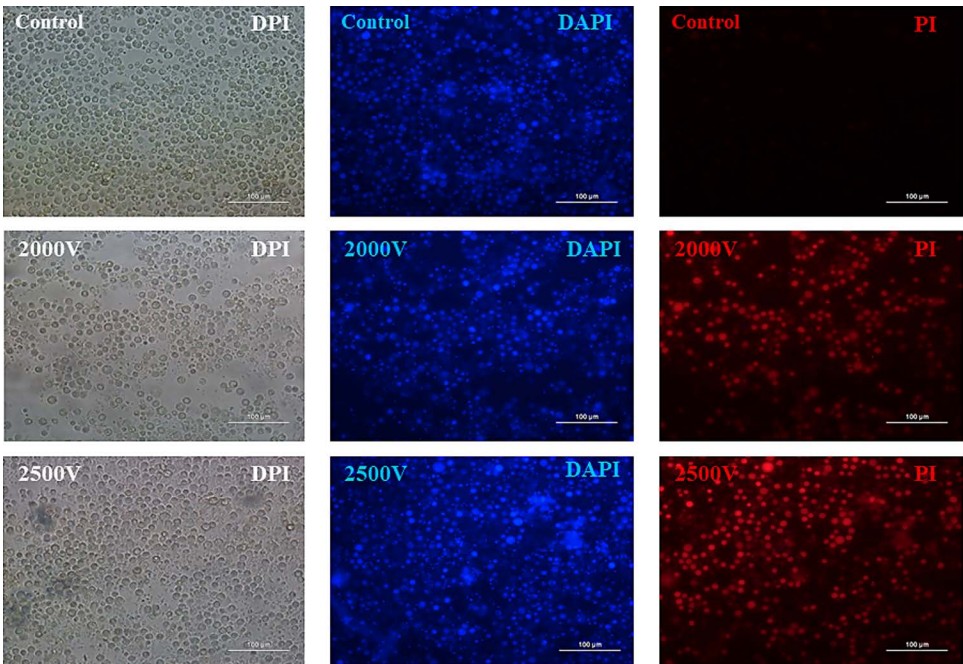

**Fig 3. *A. polyphaga* (ATCC 30461) trophozoites permeabilized with fluorescence dyes after electroporation.** Fluorescence images acquired on a Nikon Eclipse TI-U microscope of *A. polyphaga* trophozoites electroporated at 2000 and 2500 volts. DAPI: 4',6-diamidino-2-phenylindol, PI: propidium iodide (20x magnification).

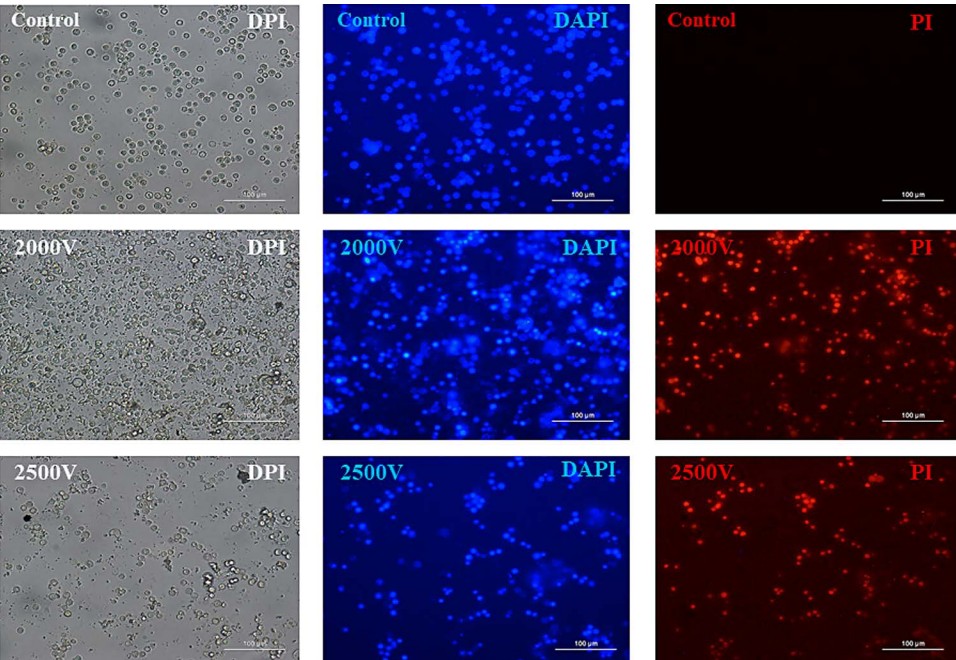

**Fig 4. *A. polyphaga* cysts (ATCC 30461) permeabilized with fluorescence dyes after electroporation.** Fluorescence images acquired under a microscope of *A. polyphaga* trophozoites electroporated at 2000 and 2500 volts. DAPI: 4',6-diamidino-2-phenylindol, PI: propidium iodide (20x magnification).

## Discussion

As far as we know, this is the first time electroporation has been proposed in ophthalmology as a treatment method for patients with *Acanthamoeba* keratitis.

Despite the advances and discoveries related to the disease, *Acanthamoeba* keratitis has remained prevalent over the years. Avoiding risk factors and obtaining an early diagnosis are still considered the most effective ways to combat the disease. However, in more advanced stages, treatment often has a long duration and high toxicity, causing considerable discomfort for patients during this period. Therefore, there is an urgent need to develop a shorter and more effective treatment [5]. This necessity led to the idea of using electroporation to combat *Acanthamoeba.*

*Acanthamoeba* is a protozoan widely distributed in the environment. It has two distinct stages in its life cycle: the trophozoite and the cyst. The trophozoite is the active form that reproduces by mitosis, exhibiting continuous movement and shape changes, and is responsible for human infection. This is particularly true in cases of eye infection associated with improper contact lens use. Under unfavorable conditions, such as nutrient and oxygen deficiency, the trophozoite gradually transforms into a cyst. Cysts have a dense double wall, composed mainly of cellulose, which provides high resistance to adverse environmental conditions for long periods while maintaining their pathogenicity. This resistance poses a significant challenge for current treatments, as it limits the ability of drugs to penetrate cyst's double wall, thereby reducing their effectiveness. If conditions become favorable again, the cyst transforms back into a trophozoite. This allows any dormant cyst, after the end of treatment, to return to its active form and restore the infection [1–5,21,22].

To improve the efficacy and shorten the treatment time of current therapeutic methods, electroceutical treatment has been proposed. The electroporation technique is a process that destabilizes the plasma membrane using electrical pulses. Depending on the intensity of the electrical pulses and the number of pulses applied, the membrane can be irreversibly ruptured, resulting in cell death. Alternatively, the membrane can be reversibly destabilized, leading to the formation of pores. This allows for the exchange of external and internal substances through the membrane, consequently enabling easier drug permeation [27–31]. Theoretically, this could enhance treatment effectiveness in cases of *Acanthamoeba* keratitis.

Thus, to improve and shorten the treatment, this study was conducted as follows: an initial inoculum (ATCC® 30461™) of $5 \times 10^6$ was prepared. The trophozoites were kept in ideal conditions of temperature and nutrients, while the cysts were placed in an environment deficient in nutrients and oxygen. A volume of 100 μL of this inoculum ($5 \times 10^5$ amoebas) per mL of electroporation buffer (285 mM sucrose, 0.7 mM MgCl2, 1 mM KCl, 10 mM HEPES, 3 mM NaOH) were added to the electroporation cuvettes. These cuvettes were then exposed to electric fields of 2. 000 and 2.500 V in different groups: group 1 (cysts and trophozoites electroporated at 2.000 volts); group 2 (cysts and trophozoites electroporated at 2.500 volts), and the control group (cysts and trophozoites without electroporation).

From then on, the first analysis was carried out to show the number of cysts and trophozoites eliminated by applying a single electrical pulse at the determined voltages. The viability analysis was performed using a Neubauer hemocytometer using 0.4% Trypan blue. This dye allows visualization of non-viable amoebas, as it is impermeable to the membrane of viable and intact cells. A significant loss of 40% of the trophozoites electroporated at 2.000 volts was observed, and 42% of those electroporated at 2.500 volts. For cysts, the loss was 13% of those electroporated at 2,000 volts and 16% of those electroporated at 2.500 volts.

After quantifying the elimination of cysts and trophozoites by the electrical pulse, a second analysis was conducted. This analysis aimed to determine, from the remaining viable cysts and trophozoites, the amount that had been permeabilized. The evaluation was based on

fluorescence intensity, where the dye DAPI (Sigma-Aldrich, USA) represents all viable cysts and trophozoites. DAPI can stain any cell with a nucleus. PI (Sigma-Aldrich, USA), which is impermeable to the wall/membrane of intact cysts and trophozoites, indicates changes in permeability; greater uptake of this fluorescence, indicates more damage to the wall/membrane. The results obtained showed that 55% of both trophozoites and cysts electroporated at 2.000 volts were permeabilized. Additionally, 59% of cysts and trophozoites electroporated at 2.500 volts were permeabilized. There was no statistically significant difference between the electroporated groups, only in comparison to the control group, where $p < 0.05$ was considered statistically significant.

Based on the research findings and the results obtained, it can be concluded that electroporation was effective in reducing *Acanthamoeba polyphaga* cysts and trophozoites at both tested voltages. This effectiveness is supported by the similarity in the levels of permeabilization observed. In the near future, lower voltages and greater quantities of pulses will be tested to achieve or even surpass the results already obtained. Further research will also explore the potential of electroporation as an adjuvant treatment to standard *Acanthamoeba* keratitis treatment. It is believed that by establishing the ideal voltages and quantities of pulses *in vitro*, we can overcome the difficulty of the drugs face in penetrating the double wall of the cyst. This may lead to the development of a shorter treatment alternative that could benefit patients.

Although electroporation is widely used for gene transfection, its application can cause adverse effects in healthy cells. This highlights that the study of electroporation has limitations. Studies indicate that an excessive electric field or prolonged permeabilization can lead to increased apoptosis in normal cells, primarily due to ionic overload and homeostatic deregulation. Additionally, the use of buffers with high conductivity can compromise cell viability. Therefore, it is essential to establish strict control of electroporation parameters in order to minimize damage and preserve cell integrity during clinical treatments [35].

Researchers have observed that *Acanthamoeba* trophozoites respond significantly to electric fields, migrating towards the cathode. This migration facilitates their access to antimicrobial agents. However, a specific and safe voltage has not yet been defined. Additionally, the inflammatory response of the cornea may vary between individuals, which could impact the effectiveness of the treatment. This suggests a need for further studies and additional tests until an efficient clinical protocol is established [36]. Even so, electric fields are already being used to treat ocular conditions such as dry eye syndrome. An example is the Rexon-Eye device, which uses low-power, high-frequency currents to stimulate metabolism and cell regeneration, restoring their function of the lacrimal and meibomian glands [37]. Given the success of this approach, electroporation is emerging as a promising option for the treatment of *Acanthamoeba* keratitis.

Another possible application being explored is electroporation as a method of cleaning contact lenses. One study revealed that the curved shape and roughness of scleral contact lenses favor the adhesion of *Acanthamoeba*, representing a significant risk of infection for lens wearers. Additionally, areas of fluid accumulation in the optical zone seem to create ideal conditions for the adhesion and proliferation of trophozoites [38]. Considering that the inappropriate use of lenses is one of the main factors associated with infection, electroporation can theoretically offer an effective solution without compromising lens integrity. By temporarily increasing the permeability of cell membranes with electrical pulses, electroporation could eliminate and/or prevent the adherence of these microorganisms. After the shock, amoebas that are not eliminated transform into cysts, which do not have the ability to adhere to surfaces.

In summary, electroporation is a versatile and promising technique that is constantly being studied and applied in medicine. It is already being used in oncology and dermatology, facilitating the introduction of therapeutic substances such as chemotherapy, genetic material,

or immunological agents for cancer treatment. In aesthetic procedures, it enhances the penetration of cosmetics into the skin, resulting in benefits such as greater firmness and reduced wrinkles. Electroporation has the potential to improve the effectiveness of conventional treatments, reduce side effects and diversify the therapeutic options available to patients. Studies highlight the safety of the technique when administered by trained professionals, minimizing damage to healthy cells. It has been proven effective in various clinical treatments. Electroporation not only optimizes therapeutic efficiency, but also expands treatment possibilities, positioning it as a safe and effective tool in contemporary medicine [32–34]. The expanded application of this technique in ophthalmology could bring substantial benefits to patients with *Acanthamoeba* keratitis.

## Acknowledgments

We would like to express our gratitude to the Center for Advanced Research on *Acanthamoeba* (CEPA), the Ophthalmology Laboratory (LOFT) and the Institute of Pharmacology and Molecular Biology (INFAR) of the research laboratories of the Federal University of São Paulo (UNIFESP), as well as Professor Sang Won Han, for their valuable contributions to the development of the project.

## Author contributions

**Data curation:** Palloma Santiago Prates Pessoa, Raphael Barcelos.

**Investigation:** Palloma Santiago Prates Pessoa.

**Methodology:** Palloma Santiago Prates Pessoa, Larissa Fagundes Pinto.

**Project administration:** Palloma Santiago Prates Pessoa.

**Supervision:** Denise de Freitas, Mauro Campos.

**Validation:** Palloma Santiago Prates Pessoa, Larissa Fagundes Pinto, Denise de Freitas, Mauro Campos.

**Writing – original draft:** Palloma Santiago Prates Pessoa.

**Writing – review & editing:** Palloma Santiago Prates Pessoa, Larissa Fagundes Pinto, Denise de Freitas, Mauro Campos.

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
