## [Decision Letter · Decision Letter 0]

11 Jun 2024

PONE-D-24-12061Effects of electroporation on Acanthamoeba

PolyphagaPLOS ONE

Dear Dr. Pessoa,

Thank you for submitting your manuscript to PLOS ONE. After careful consideration, we feel that it has merit but does not fully meet PLOS ONE’s publication criteria as it currently stands. Therefore, we invite you to submit a revised version of the manuscript that addresses the points raised during the review process.

We look forward to receiving your revised manuscript.

Kind regards,

Alireza Badirzadeh

Academic Editor

PLOS ONE

Journal Requirements:

Reviewers' comments:

Reviewer's Responses to Questions

**Comments to the Author**

1. Is the manuscript technically sound, and do the data support the conclusions?

Reviewer #1: Partly

Reviewer #2: Yes

2. Has the statistical analysis been performed appropriately and rigorously?

Reviewer #1: I Don't Know

Reviewer #2: Yes

3. Have the authors made all data underlying the findings in their manuscript fully available?

Reviewer #1: Yes

Reviewer #2: Yes

4. Is the manuscript presented in an intelligible fashion and written in standard English?

Reviewer #1: Yes

Reviewer #2: Yes

5. Review Comments to the Author

**Reviewer #1:**  1. This article has previously been presented at the following congress: CAMPOS, M. S. Q. ; FREITAS, D. ; BARCELOS, R. ; PINTO, L. F. ; PESSOA, P. S. P. . EFFECTS OF ELECTROPORATION ON Acanthamoeba Polyphaga. 2023. (Apresentação de Trabalho/Congresso). Based on the link: https://sbpz.org.br/wp-content/uploads/2023/09/livro-inteiro-04set.pdf

2. There are some similarities in this article with other published articles.

**Reviewer #2:**  Abstract section

1. Lack of information regarding the methods used for performing electroporation on cysts and trophozoites.

2. Insufficient details about the presented results, including data interpretation and statistical distributions.

3. Need for technical enhancement in presenting the results in a manner that allows for more precise analysis and comprehensive interpretation.

4. Absence of sufficient information regarding the significance of the obtained results and the necessity for clearer articulation of this significance.

5. Requirement for delineating limitations and suggestions for future research in the abstract.

Introduction section

1. Line 50: "genus Acanthamoeba spp." should be "genus Acanthamoeba" as "spp." indicates multiple species, which is not necessary here.

2. Line 54: Instead of "evolutionary forms," use "life cycle stages" for more accurate terminology.

3. Line 57: Change "These are latent and resistant forms, allowing them to remain and spread in the environment, as well as becoming chronic in cases of eye disease" to "These are dormant and resistant forms, enabling them to persist and spread in the environment, and to cause chronic infections in cases of eye disease."

4. Line 62: Replace "23 different genotypes (T1-T23)" with "23 distinct genotypes (T1-T23)."

5. Line 64: "in humans are relatively uncommon" should be "are relatively uncommon in humans."

6. Line 69: Change "which can easily be confused with other ocular infections" to "often confused with other ocular infections."

7. Line 70: Replace "expanding to severe cases with more specific symptoms, severe and painful ulcers in the corneal region, which can spread to the inner layers of the eyeball" with "progressing to severe cases with more specific symptoms, including severe and painful corneal ulcers that can extend to the inner layers of the eyeball."

8. Line 74: "throughout the therapeutic process" should be "during the treatment process."

9. Line 85: "making it less time-consuming for patients" should be "reducing the treatment duration for patients."

10. Line 88: "applying electric fields" should be "application of electric fields."

11. Line 91: Change "configured to form hydrophilic pores" to "adjusted to form hydrophilic pores."

12. Line 94: "and then closes, after which cellular equilibrium is re-established" should be "and then closes, allowing cellular equilibrium to be re-established."

13. Line 95: Change "With more exacerbated pulse parameters, the electroporation process becomes irreversible, as the cell is unable to restructure itself" to "With higher pulse parameters, electroporation becomes irreversible, preventing the cell from restructuring."

14. Ensure the reference to "Pussard and Pons classification, 1977" is accurate and from a reliable source.

15. Provide more details about the safety and efficacy of the electroporation method.

16. Mentioning the potential use of electroporation to facilitate treatment should be backed by supporting studies and scientific research.

• The introduction is quite lengthy. Consider summarizing some sections and focusing on the most critical points to maintain reader engagement and clarity.

Methodology Section:

- The methodology section is generally well-organized, but there are some areas where clarity and detail can be improved. Additionally, consistent formatting and correct scientific terminology are essential.

1. Line 105: "A standardized Acanthamoeba strain was used in this study: Acanthamoeba Polyphaga (American Type Culture Collection, ATCC® 30461TM) obtained from a corneal scrape of a CA case in Texas, 1973."

• Change "CA case" to "keratitis case."

• Consider rephrasing for clarity: "A standardized strain of Acanthamoeba polyphaga (ATCC® 30461TM) was used in this study, obtained from a corneal scrape of a keratitis case in Texas, 1973."

2. Line 109: "The ATCC 30461 isolate was grown on non-nutrient agar medium supplemented with heat-inactivated E. coli (DH5α) at 25ºC."

• Remove the line break before "The ATCC 30461 isolate."

• "heat-inactivated" should be hyphenated.

3. Line 110: "After these strains had been disentangled, the non-nutrient agar plate was washed with transport saline and the wash was centrifuged at 2000 RPM for 15 minutes."

• "disentangled" is not the correct term. Use "harvested."

• Specify the exact centrifuge conditions: "washed with transport saline and centrifuged at 2000 RPM for 15 minutes."

4. Line 112: "After centrifugation, the supernatant was discarded and the amoeba pellet re suspended in 1mL of culture medium, which was then inoculated into a 25 cm2 culture flask containing 4mL of culture medium and incubated in an oven at 25ºC."

• "re suspended" should be "resuspended."

• Clarify "culture medium" to specify the type.

5. Line 114: "To obtain the trophozoite forms, the isolate was grown in PYG (proteose-peptone - yeast extract - glucose) culture medium and, for the cystic forms, it was grown in Neff encystment medium."

• Clarify: "To obtain the trophozoite forms, the isolate was grown in PYG (proteose-peptone - yeast extract - glucose) culture medium. For cystic forms, it was grown in Neff's encystment medium."

6. Line 119: "Trophozoites and cysts of the ATCC 30461 isolate were counted in a Neubauer chamber and their concentration was adjusted to obtain an initial inoculum of 5x106 trophozoites/cysts per mL of electroporation buffer (285mM sacarose, 0,7 MgCl2, 1mM KlC, 10mM HEPES, 3Mm NaOH)."

• Correct "sacarose" to "sucrose."

• Correct "0,7 MgCl2" to "0.7 mM MgCl2."

• Correct "KlC" to "KCl."

• Correct "3Mm NaOH" to "3 mM NaOH."

7. Line 122: "Thus, 100μL of this inoculum (5x105 amoebas) were added to the electroporation cuvettes exposed to an electric field of 2000 and 2500V."

• Clarify: "Thus, 100μL of this inoculum (5x105 amoebas) was added to the electroporation cuvettes, which were then exposed to an electric field of 2000 or 2500V."

8. Line 123: "Subsequently, methods for detecting electroporation involving fluorescence, described in detail below in item “Electroporation detection”, were carried out to standardize and evaluate the best voltage to be applied to Acanthamoeba cysts and trophozoites."

• "Subsequently, fluorescence-based methods for detecting electroporation were carried out as described below in the “Electroporation detection” section, to standardize and evaluate the optimal voltage for Acanthamoeba cysts and trophozoites."

9. Line 129: "The in vitro experimental model was divided into the following groups: cysts and trophozoites controls (without electroporation); group 1: cysts and trophozoites electroporated at 2000 volts; group 2: cysts and trophozoites electroporated at 2500 volts."

• Use bullet points or a clearer list format for readability:

Controls (cysts and trophozoites without electroporation)

Group 1: cysts and trophozoites electroporated at 2000 volts

Group 2: cysts and trophozoites electroporated at 2500 volts

10. Line 138: "To assess the loss of cysts and trophozoites from the ATCC 30461 isolate resulting only from the shock applied, a separate analysis was first carried out counting the number of amoebas in each group before and after electroporation using Trypan blue 0.4% in a Neubauer hemocytometer."

• "resulting only from the shock applied" can be simplified: "To assess the loss of cysts and trophozoites due to the applied shock, a separate analysis was performed. Amoebas in each group were counted before and after electroporation using 0.4% Trypan blue in a Neubauer hemocytometer."

11. Line 146: "Subsequently, another analysis was carried out using the fluorescence intensity method. The electroporated cysts and trophozoites were plated in 96-well plates and stained with the fluorescent dye propidium iodide (PI; Sigma-Aldrich, USA)."

• Simplify: "Another analysis was performed using fluorescence intensity. Electroporated cysts and trophozoites were plated in 96-well plates and stained with propidium iodide (PI; Sigma-Aldrich, USA)."

12. Line 152: "The fluorescent dye 4',6-diamidino-2-phenylindol (DAPI; Sigma-Aldrich, USA), acts as a marker, coloring any cell with a blue nucleus."

• Correct spelling: "4',6-diamidino-2-phenylindole (DAPI; Sigma-Aldrich, USA), acts as a marker, staining the nucleus of any cell blue."

13. Line 167: "Statistical analyses were carried out using the GraphPad Prism 8.0 program. The two-way ANOVA test was used for multiple comparisons between groups. Values of p < 0.05 were considered statistically significant."

• Minor correction: "Statistical analyses were carried out using GraphPad Prism 8.0. A two-way ANOVA test was used for multiple comparisons between groups. Values of p < 0.05 were considered statistically significant."

In discussion section

1. Some concepts need to be explained in simpler terms to enhance reader comprehension.

2. Provide more specific numbers and detailed information about the experimental setup, methodology, and results. Additionally, ensure consistency in referencing previous studies.

3. Elaborate on certain aspects of the results and methods to provide a clearer understanding, especially regarding any discrepancies observed in the experiments.

4. Offer a more detailed interpretation of the results, including comparisons with similar studies and a discussion of potential implications and limitations.

5. Discuss the generalizability of the findings beyond the laboratory setting and their applicability in real-world clinical scenarios.

Overall, improvements in expressing concepts, more precise information about results, additional explanations, and a more logical interpretation of the results can enhance the paper.

As a reviewer of the manuscript, I recommend addressing the low quality of the images and graphs. It is essential that these visual elements be of higher quality to effectively convey the research findings. Therefore, I strongly advise the authors to improve the resolution and clarity of the images and graphs before finalizing the manuscript for publication. This enhancement will significantly enhance the overall presentation and comprehension of the study.

6. PLOS authors have the option to publish the peer review history of their article (what does this mean? ). If published, this will include your full peer review and any attached files.

**Do you want your identity to be public for this peer review?** For information about this choice, including consent withdrawal, please see our Privacy Policy .

Reviewer #1: No

Reviewer #2: **Yes: ** Ahmad Hosseini-Safa

---

## [Author Response · Author response to Decision Letter 0]

12 Jul 2024

- After review, we believe that the manuscript fully complies with the PLOS ONE guidelines.

Please confirm at this time whether or not your submission contains all raw data required to replicate the results of your study. Authors must share the “minimal data set” for their submission.

- We confirm that all the minimum information necessary to replicate our study has been included in the manuscript. No data from the analysis has been omitted, and all experimental procedures are recorded in the laboratory minutes. All relevant data for analysis are available as defined by the minimum data set required for replication according to PLOS guidelines.

Reviewer #1:

1. This article has previously been presented at the following congress: CAMPOS, M. S. Q. ; FREITAS, D. ; BARCELOS, R. ; PINTO, L. F. ; PESSOA, P. S. P. . EFFECTS OF ELECTROPORATION ON Acanthamoeba Polyphaga. 2023. (Apresentação de Trabalho/Congresso). Based on the link: https://sbpz.org.br/wp-content/uploads/2023/09/livro-inteiro-04set.pdf

1.1. During the course of the study, it was partially presented at the XXXVI Annual Meeting of the Brazilian Society of Protozoology, 2023. This is the first full disclosure of the study.

2. There are some similarities in this article with other published articles.

2.1. To the best of our knowledge, this is the first study designed for quantitative evaluation of the viability and permeabilization of Acanthamoeba polyphaga using the electroporation technique.

Reviewer #2:

Abstract section

1. Lack of information regarding the methods used for performing electroporation on cysts and trophozoites.

1.1. The abstract has been revised to include detailed information on the method used to carry out electroporation on cysts and trophozoites, as requested.

2. Insufficient details about the presented results, including data interpretation and statistical distributions.

2.1. The section on the results has been improved to provide a better interpretation of the data, including statistical distributions.

3. Need for technical enhancement in presenting the results in a manner that allows for more precise analysis and comprehensive interpretation.

3.1. The presentation of the results has been modified to allow for more precise analysis and comprehensive interpretation.

4. Absence of sufficient information regarding the significance of the obtained results and the necessity for clearer articulation of this significance.

4.1. It is believed that the summary now more clearly articulates the significance of the results obtained, explaining their relevance.

5. Requirement for delineating limitations and suggestions for future research in the abstract.

5.1. The mention of future research has been removed. The conclusion now focuses on the interpretation that the research suggests the possibility of a complementary treatment to the standard treatment.

* In general, the abstract has been reformulated to meet all the requirements, respecting the established character limit.

Introduction section

1. Line 50: "genus Acanthamoeba spp." should be "genus Acanthamoeba" as "spp." indicates multiple species, which is not necessary here.

1.1 "spp." has been removed

2. Line 54: Instead of "evolutionary forms," use "life cycle stages" for more accurate terminology.

2.1 The modification has been made.

3. Line 57: Change "These are latent and resistant forms, allowing them to remain and spread in the environment, as well as becoming chronic in cases of eye disease" to "These are dormant and resistant forms, enabling them to persist and spread in the environment, and to cause chronic infections in cases of eye disease."

3.1 The modification has been made.

4. Line 62: Replace "23 different genotypes (T1-T23)" with "23 distinct genotypes (T1-T23)."

4.1 The modification has been made.

5. Line 64: "in humans are relatively uncommon" should be "are relatively uncommon in humans."

5.1 The modification has been made.

6. Line 69: Change "which can easily be confused with other ocular infections" to "often confused with other ocular infections."

6.1 The modification has been made.

7. Line 70: Replace "expanding to severe cases with more specific symptoms, severe and painful ulcers in the corneal region, which can spread to the inner layers of the eyeball" with "progressing to severe cases with more specific symptoms, including severe and painful corneal ulcers that can extend to the inner layers of the eyeball."

7.1 The modification has been made.

8. Line 74: "throughout the therapeutic process" should be "during the treatment process."

8.1 The modification has been made.

9. Line 85: "making it less time-consuming for patients" should be "reducing the treatment duration for patients."

9.1 The modification has been made.

10. Line 88: "applying electric fields" should be "application of electric fields."

10.1 The modification has been made.

11. Line 91: Change "configured to form hydrophilic pores" to "adjusted to form hydrophilic pores."

11.1 The modification has been made.

12. Line 94: "and then closes, after which cellular equilibrium is re-established" should be "and then closes, allowing cellular equilibrium to be re-established."

12.1 The modification has been made.

13. Line 95: Change "With more exacerbated pulse parameters, the electroporation process becomes irreversible, as the cell is unable to restructure itself" to "With higher pulse parameters, electroporation becomes irreversible, preventing the cell from restructuring".

13.1 The modification has been made.

14. Ensure the reference to "Pussard and Pons classification, 1977" is accurate and from a reliable source.

14.1 We apologize for the mistake; we have corrected it by adding the correct reference to the text.

“Putaporntip C, Kuamsab N, Nuprasert W, Rojrung R, Pattanawong U, Tia T, et al. Analysis of Acanthamoeba genotypes from public freshwater sources in Thailand reveals a new genotype, T23 Acanthamoeba bangkokensis sp. nov. Sci Rep. 2021;11(1):17290.”

15. Provide more details about the safety and efficacy of the electroporation method.

15.1 As far as we know, this is the first time that electroporation has been proposed in the field of ophthalmology. Therefore, in the text, we address the safety and efficacy of the method based on consolidated research in other areas of medicine.

16. Mentioning the potential use of electroporation to facilitate treatment should be backed by supporting studies and scientific research.

16.1 We draw on the successful application of electroporation in other areas of medicine to suggest a possible adjunctive treatment to the standard treatment for Acanthamoeba keratitis.

“Bonadies A, Bertozzi E, Cristiani R, Govoni F, Migliano E. Electrochemotherapy in Skin Malignancies of Head and Neck Cancer Patients: Clinical Efficacy and Aesthetic Benefits. Acta Dermato Venereologica. 2019;99(13):1246–52.”

“Zhang W, Wang W, Chai W, Luo X, Li J, Shi J, et al. Breast tissue ablation with irreversible electroporation in rabbits: A safety and feasibility study. Scarfi MR, editor. PLOS ONE [Internet]. 2017 Jul 21 [cited 2019 Dec 19];12(7):e0181555.”

“Jean-Pierre Tasu, Tougeron D, Marie-Pierre Rols. Irreversible electroporation and electrochemotherapy in oncology: State of the art. 2022 Nov 1;103(11):499–509.”

• The introduction is quite lengthy. Consider summarizing some sections and focusing on the most critical points to maintain reader engagement and clarity.

- After re-examining the introduction, we decided not to reduce it any further, as we believe that all the information it contains is of fundamental importance for the reader's understanding.

Methodology Section:

- The methodology section is generally well-organized, but there are some areas where clarity and detail can be improved. Additionally, consistent formatting and correct scientific terminology are essential.

1. Line 105: "A standardized Acanthamoeba strain was used in this study: Acanthamoeba Polyphaga (American Type Culture Collection, ATCC® 30461TM) obtained from a corneal scrape of a CA case in Texas, 1973."

• Change "CA case" to "keratitis case."

• Consider rephrasing for clarity: "A standardized strain of Acanthamoeba polyphaga (ATCC® 30461TM) was used in this study, obtained from a corneal scrape of a keratitis case in Texas, 1973."

1.1 The modifications have been made.

2. Line 109: "The ATCC 30461 isolate was grown on non-nutrient agar medium supplemented with heat-inactivated E. coli (DH5α) at 25ºC."

• Remove the line break before "The ATCC 30461 isolate."

• "heat-inactivated" should be hyphenated.

2.1 The modifications have been made.

3. Line 110: "After these strains had been disentangled, the non-nutrient agar plate was washed with transport saline and the wash was centrifuged at 2000 RPM for 15 minutes."

• "disentangled" is not the correct term. Use "harvested."

• Specify the exact centrifuge conditions: "washed with transport saline and centrifuged at 2000 RPM for 15 minutes."

3.1 The modifications have been made.

4. Line 112: "After centrifugation, the supernatant was discarded and the amoeba pellet re suspended in 1mL of culture medium, which was then inoculated into a 25 cm2 culture flask containing 4mL of culture medium and incubated in an oven at 25ºC."

• "re suspended" should be "resuspended."

• Clarify "culture medium" to specify the type.

4.1 The modification has been made.

5. Line 114: "To obtain the trophozoite forms, the isolate was grown in PYG (proteose-peptone - yeast extract - glucose) culture medium and, for the cystic forms, it was grown in Neff encystment medium."

• Clarify: "To obtain the trophozoite forms, the isolate was grown in PYG (proteose-peptone - yeast extract - glucose) culture medium. For cystic forms, it was grown in Neff's encystment medium."

5.1 The modification has been made.

6. Line 119: "Trophozoites and cysts of the ATCC 30461 isolate were counted in a Neubauer chamber and their concentration was adjusted to obtain an initial inoculum of 5x106 trophozoites/cysts per mL of electroporation buffer (285mM sacarose, 0,7 MgCl2, 1mM KlC, 10mM HEPES, 3Mm NaOH)."

• Correct "sacarose" to "sucrose."

• Correct "0,7 MgCl2" to "0.7 mM MgCl2."

• Correct "KlC" to "KCl."

• Correct "3Mm NaOH" to "3 mM NaOH.""

6.1 The modifications have been made.

7. Line 122: "Thus, 100μL of this inoculum (5x105 amoebas) were added to the electroporation cuvettes exposed to an electric field of 2000 and 2500V."

• Clarify: "Thus, 100μL of this inoculum (5x105 amoebas) was added to the electroporation cuvettes, which were then exposed to an electric field of 2000 or 2500V."

7.1 The modification has been made.

8. Line 123: "Subsequently, methods for detecting electroporation involving fluorescence, described in detail below in item “Electroporation detection”, were carried out to standardize and evaluate the best voltage to be applied to Acanthamoeba cysts and trophozoites."

• Subsequently, fluorescence-based methods for detecting electroporation were carried out as described below in the “Electroporation detection” section, to standardize and evaluate the optimal voltage for Acanthamoeba cysts and trophozoites."

8.1 The modification has been made.

9. Line 129: "The in vitro experimental model was divided into the following groups: cysts and trophozoites controls (without electroporation); group 1: cysts and trophozoites electroporated at 2000 volts; group 2: cysts and trophozoites electroporated at 2500 volts."

• Use bullet points or a clearer list format for readability:

a. Controls (cysts and trophozoites without electroporation)

b. Group 1: cysts and trophozoites electroporated at 2000 volts

c. Group 2: cysts and trophozoites electroporated at 2500 volts

9.1 The modification has been made.

10. Line 138: "To assess the loss of cysts and trophozoites from the ATCC 30461 isolate resulting only from the shock applied, a separate analysis was first carried out counting the number of amoebas in each group before and after electroporation using Trypan blue 0.4% in a Neubauer hemocytometer."

• "resulting only from the shock applied" can be simplified: "To assess the loss of cysts and trophozoites due to the applied shock, a separate analysis was performed. Amoebas in each group were counted before and after electroporation using 0.4% Trypan blue in a Neubauer hemocytometer."

10.1 The modification has been made.

11. Line 146: "Subsequently, another analysis was carried out using the fluorescence intensity method. The electroporated cysts and trophozoites were plated in 96-well plates and stained with the fluorescent dye propidium iodide (PI; Sigma-Aldrich, USA)."

• Simplify: "Another analysis was performed using fluorescence intensity. Electroporated cysts and trophozoites were plated in 96-well plates and stained with propidium iodide (PI; Sigma-Aldrich, USA)."

11.1 The modification has been made.

12. Line 152: "The fluorescent dye 4',6-diamidino-2-phenylindol (DAPI; Sigma-Aldrich, USA), acts as a marker, coloring any cell with a blue nucleus."

• Correct spelling: "4',6-diamidino-2-phenylindole (DAPI; Sigma-Aldrich, USA), acts as a marker, staining the nucleus of any cell blue."

12.1 The modification has been made.

13. Line 167: "Statistical analyses were carried out using the GraphPad Prism 8.0 program. The two-way ANOVA test was used for multiple comparisons between groups. Values of p < 0.05 were considered statistically significant."

• Minor correction: "Statistical analyses were carried out using GraphPad Prism 8.0. A two-way ANOVA test was used for multiple comparisons between groups. Values of p < 0.05 were considered statistically significant."

13.1 The modification has been made.

Discussion Section:

1. Some concepts need to be explained in simpler terms to enhance reader comprehension.

1.1 We have revised the section to explain some concepts in simpler terms in order to improve the reader's understanding. We have done our best to make the language more accessible without compromising technical precision.

2. Provide more specific numbers and detailed information about the experimental setup, methodology, and results. Additionally, ensure consistency in referencing previous studies.

2.1 We provide more detailed information on the experimental setup, methodology and results. We ensure that the sources cited are clearly identified and correctly referenced.

3. Elaborate on certain aspects of the results and methods to provide a clearer understanding, especially regarding any discrepancies observed in the experiments.

3.1 We have elaborated on aspects of the results and methods to provide a clearer understanding, especially in relation to any discrepancies observed in the experiments.

4. Offer a more detailed interpretation of the results, including comparisons with similar studies and a discussion of potential implications and limitations.

4.1 We provide a more detailed interpretation of the results, including comparisons with studies that apply the technique in other areas of medicine, with the aim of providing a more in-depth and contextualized analysis.

5. Discuss the generalizability of the findings beyond the laboratory setting and their applicability in real-world clinical scenarios.

5.1 We include in the discussion the importance of the results beyond the laboratory environment in addition to their possible applicability in practical clinical contexts in the real world and the necessary considerations for this application.

Overall, improvements in expressing concepts, more precise information about results, additional explanations, and a more logical interpretation of the results can enhance the paper.

As a reviewer of the manuscript, I recommend addressing the low quality of the images and graphs. It is essential that these visual elements be of higher quality to effectively convey the research

---

## [Decision Letter · Decision Letter 1]

7 Oct 2024

PONE-D-24-12061R1Effects of electroporation on Acanthamoeba

polyphagaPLOS ONE

Dear Dr. Pessoa,

Thank you for submitting your manuscript to PLOS ONE. After careful consideration, we feel that it has merit but does not fully meet PLOS ONE’s publication criteria as it currently stands. Therefore, we invite you to submit a revised version of the manuscript that addresses the points raised during the review process.

We look forward to receiving your revised manuscript.

Kind regards,

Alireza Badirzadeh

Academic Editor

PLOS ONE

Journal Requirements:

Reviewers' comments:

Reviewer's Responses to Questions

**Comments to the Author**

1. If the authors have adequately addressed your comments raised in a previous round of review and you feel that this manuscript is now acceptable for publication, you may indicate that here to bypass the “Comments to the Author” section, enter your conflict of interest statement in the “Confidential to Editor” section, and submit your "Accept" recommendation.

Reviewer #1: (No Response)

Reviewer #2: All comments have been addressed

2. Is the manuscript technically sound, and do the data support the conclusions?

Reviewer #1: Yes

Reviewer #2: Yes

3. Has the statistical analysis been performed appropriately and rigorously?

Reviewer #1: I Don't Know

Reviewer #2: Yes

4. Have the authors made all data underlying the findings in their manuscript fully available?

Reviewer #1: Yes

Reviewer #2: Yes

5. Is the manuscript presented in an intelligible fashion and written in standard English?

Reviewer #1: Yes

Reviewer #2: Yes

6. Review Comments to the Author

Reviewer #1: 1. Since a part of this article has already been presented in the Congress, if it does not conflict with the policies of this journal, it can be printed.

2. It is better to write the sample storage and transfer method.

Reviewer #2: 1. Original Research:

The study presents novel research focused on using electroporation to treat Acanthamoeba polyphaga cysts and trophozoites, which addresses an important area in ophthalmology.

The originality of the topic, especially the use of electroporation as a potential adjunct therapy, is evident and relevant. It appears to have been presented in conferences before, but the authors clarify that this is the first full disclosure of the study.

2. Unpublished Results:

The results do not seem to have been previously published elsewhere, except for a presentation at a conference, which is common practice.

3. Technical Standards and Clarity of Methods:

The methods are described with sufficient technical detail, allowing replication of the experiment.

However, clarity in some areas of the methodology could be improved. For example, the explanation of the electroporation setup, such as voltage pulse durations and the number of pulses, could be expanded for clarity.

4. Appropriate Conclusions:

The conclusions are consistent with the data, particularly regarding the effectiveness of electroporation for Acanthamoeba polyphaga.

However, the conclusions might benefit from further elaboration on the clinical implications of the findings. While the results are promising, discussing potential hurdles in clinical application (such as safety or cost-effectiveness) would add depth.

5. Presentation and Language:

The article is mostly well-written and intelligible, but there are some minor language issues that can be improved for better readability. For instance, sentence structure could be simplified in sections like the discussion. Some scientific terminology is overly technical for a general reader, so clarification may be needed in specific areas.

Spelling and grammatical issues: There are a few minor errors, such as "re suspended" (should be "resuspended"), "sacarose" (should be "sucrose"), and "permaebilization" (should be "permeabilization").

6. Ethical Standards and Research Integrity:

The study adheres to ethical standards, although there is no explicit mention of ethics approval, which should be included if necessary for any animal or human-related procedures (although this may not apply here).

There are no competing interests or financial disclosures that affect the research.

7. Data Availability and Reporting Standards:

The authors have confirmed that all data necessary to replicate the study are included, and appropriate figures and tables support the data.

Recommendations for Improvement:

Clarification in Methods: Improve the explanation of the electroporation parameters, including the exact number of pulses and pulse duration.

Language Refinement: Fix minor grammatical and structural issues to enhance clarity.

Further Interpretation: Expand on the implications of the findings in a clinical context, and discuss potential limitations or challenges in applying electroporation for AK treatment.

Ethics Approval: Ensure that any necessary ethical statements are included, if applicable.

7. PLOS authors have the option to publish the peer review history of their article (what does this mean? ). If published, this will include your full peer review and any attached files.

**Do you want your identity to be public for this peer review?** For information about this choice, including consent withdrawal, please see our Privacy Policy .

Reviewer #1: **Yes: ** Fares Bahrami

Reviewer #2: No

---

## [Author Response · Author response to Decision Letter 1]

12 Nov 2024

Journal Requirements:

1. Please review your reference list to ensure that it is complete and correct. If you have cited papers that have been retracted, please include the rationale for doing so in the manuscript text, or remove these references and replace them with relevant current references. Any changes to the reference list should be mentioned in the rebuttal letter that accompanies your revised manuscript. If you need to cite a retracted article, indicate the article’s retracted status in the References list and also include a citation and full reference for the retraction notice. After review, we believe that the manuscript fully complies with the PLOS ONE guidelines.

- We confirm that all references have been carefully reviewed to ensure that they comply with PLOS ONE standards. The added references since the first review are the following:

“32. Bonadies A, Bertozzi E, Cristiani R, Govoni F, Migliano E. Electrochemotherapy in Skin Malignancies of Head and Neck Cancer Patients: Clinical Efficacy and Aesthetic Benefits. Acta Dermato Venereol. 2019;99(13):1246–52.”

“33. Zhang W, Wang W, Chai W, Luo X, Li J, Shi J, et al. Breast tissue ablation with irreversible electroporation in rabbits: A safety and feasibility study. Scarfi MR, editor. PLOS ONE. 2017;12(7):e0181555.”

“34. Jean-Pierre Tasu, Tougeron D, Rols MP. Irreversible electroporation and electrochemotherapy in oncology: State of the art. Bull Cancer. 2022; 103(11):499–509.

35. Sherba JJ, Hogquist S, Lin H, Shan JW, Shreiber DI, Zahn JD. The effects of electroporation buffer composition on cell viability and electro-transfection efficiency. Sci Rep. 2020;10:3053.

36. Carvalho FR. Clinical implications of Acanthamoeba affinity for electric fields. Invest Ophthalmol Vis Sci. 2013;54(6):4234.

37. Resono Ophthalmic Srl. Rexon-Eye® (QMR® generator). Sandrigo, Italy. Available from: www.resono.it.

38. Pinto LF, Rott MB, Barsch MCS, Rocchetti TT, Yu MCZ, Sant'Ana VP, Gatti ÍMV, Rocha LL, Hofling-Lima AL, de Freitas D. Adhesion of Acanthamoeba on Scleral Contact Lenses According to Lens Shape. Invest Ophthalmol Vis Sci. 2024 May 1;65(5):4.”

Comments to the Author

1. If the authors have adequately addressed your comments raised in a previous round of review and you feel that this manuscript is now acceptable for publication, you may indicate that here to bypass the “Comments to the Author” section, enter your conflict of interest statement in the “Confidential to Editor” section, and submit your "Accept" recommendation.

Reviewer #1: (No response)

Reviewer #2: manuscript revised accordingly

2. Is the manuscript technically sound, and do the data support the conclusions?

Reviewer #1: Yes

Reviewer #2: Yes

3. Has the statistical analysis been performed appropriately and rigorously?

Reviewer #1: I don’t know

Reviewer #2: Yes

4. Have the authors made all data underlying the findings in their manuscript fully available?

Reviewer #1: Yes

Reviewer #2: Yes

5. Is the manuscript presented in an intelligible fashion and written in standard English?

Reviewer #1: Yes

Reviewer #2: Yes

6. Review Comments to the Author

We confirm that all the requested corrections have been made. The description of the electroporation parameters has been improved, including the number and duration of pulses, as recommended. We have also revised the manuscript to correct grammatical errors and improve clarity. Regarding ethical approval, we would like to clarify that the study did not involve animal procedures or human subjects, so an ethics statement is not required. We thank you for your careful review and are at your disposal for any further clarification.

Reviewer #1:

1. Since a part of this article has already been presented in the Congress, if it does not conflict with the policies of this journal, it can be printed.

1.1. We would like to stress that this is the first time the results have been published in full. Although part of it has been presented at a conference, the complete and updated content is being submitted for the first time for publication in the journal, without any conflict with editorial policies.

2. It is better to write the sample storage and transfer method.

2.1. We appreciate your suggestion and apologize, but we have not been able to identify exactly what is being proposed.

If you are referring to the storage of the ATCC® 30461TM sample prior to culture, the sample is kept in an ultra-freezer at -80°C, using a freezing solution containing 7.5% DMSO to preserve the viability of the cells. Subsequently, these samples are sent for culture according to the procedures described in the topic “Acanthamoeba Polyphaga Culture”, after which the experiments are conducted.

If you have any questions about handling after the electroporation process, the samples are kept at 4°C in 96-well plates for checking and control purposes only, but please note that a new batch of amoebas is removed from the culture for each experiment.

Reviewer #2:

1. Original Research:

The study presents novel research focused on using electroporation to treat Acanthamoeba polyphaga cysts and trophozoites, which addresses an important area in ophthalmology.

The originality of the topic, especially the use of electroporation as a potential adjunct therapy, is evident and relevant. It appears to have been presented in conferences before, but the authors clarify that this is the first full disclosure of the study.

1.1. Thank you for your comments on the relevance and originality of the study. In fact, the work presents a new approach using electroporation in the treatment of Acanthamoeba polyphaga, with great therapeutic potential in ophthalmology. Although the subject has been briefly presented at conferences, we would like to point out that this is the first time that the complete and detailed results of the study have been published. We believe that this publication offers a significant contribution to the field and hope that it will generate further discussion and advances in the treatment of Acanthamoeba keratitis.

2. Unpublished Results:

The results do not seem to have been previously published elsewhere, except for a presentation at a conference, which is common practice.

2.1. Thank you for your comment. We confirm that this is the first complete and detailed publication of the study.

3. Technical Standards and Clarity of Methods:

The methods are described with sufficient technical detail, allowing replication of the experiment.

However, clarity in some areas of the methodology could be improved. For example, the explanation of the electroporation setup, such as voltage pulse durations and the number of pulses, could be expanded for clarity.

3.1. We appreciate your input and have made improvements to enhance clarity. Specifically, we included the detail that the amoebas were exposed to an electric field of 2000 and 2500 V in a single pulse of approximately 1.4 seconds in the "Electroporation" section.

4. Appropriate Conclusions:

The conclusions are consistent with the data, particularly regarding the effectiveness of electroporation for Acanthamoeba polyphaga.

However, the conclusions might benefit from further elaboration on the clinical implications of the findings. While the results are promising, discussing potential hurdles in clinical application (such as safety or cost-effectiveness) would add depth.

4.1. We appreciate your suggestion and recognize that the conclusions could be enriched with a more detailed discussion of the clinical implications of the results. We have addressed potential challenges in clinical application, such as safety issues, which has certainly added more depth and sheds more light on the prospects for practical implementation. Added paragraphs follow:

- “Although electroporation is widely used for gene transfection, its application can cause adverse effects in healthy cells. This highlights that the study of electroporation has limitations. Studies indicate that an excessive electric field or prolonged permeabilization can lead to increased apoptosis in normal cells, primarily due to ionic overload and homeostatic deregulation. Additionally, the use of buffers with high conductivity can compromise cell viability. Therefore, it is essential to establish strict control of electroporation parameters in order to minimize damage and preserve cell integrity during clinical treatments (35).

- Researchers have observed that Acanthamoeba trophozoites respond significantly to electric fields, migrating towards the cathode. This migration facilitates their access to antimicrobial agents. However, a specific and safe voltage has not yet been defined. Additionally, the inflammatory response of the cornea may vary between individuals, which could impact the effectiveness of the treatment. This suggests a need for further studies and additional tests until an efficient clinical protocol is established (36). Even so, electric fields are already being used to treat ocular conditions such as dry eye syndrome. An example is the Rexon-Eye device, which uses low-power, high-frequency currents to stimulate metabolism and cell regeneration, restoring their function of the lacrimal and meibomian glands (37). Given the success of this approach, electroporation is emerging as a promising option for the treatment of Acanthamoeba keratitis.

- Another possible application being explored is electroporation as a method of cleaning contact lenses. One study revealed that the curved shape and roughness of scleral contact lenses favor the adhesion of Acanthamoeba, representing a significant risk of infection for lens wearers. Additionally, areas of fluid accumulation in the optical zone seem to create ideal conditions for the adhesion and proliferation of trophozoites (38). Considering that the inappropriate use of lenses is one of the main factors associated with infection, electroporation can theoretically offer an effective solution without compromising lens integrity. By temporarily increasing the permeability of cell membranes with electrical pulses, electroporation could eliminate and/or prevent the adherence of these microorganisms. After the shock, amoebas that are not eliminated transform into cysts, which do not have the ability to adhere to surfaces.”

5. Presentation and Language:

The article is mostly well-written and intelligible, but there are some minor language issues that can be improved for better readability. For instance, sentence structure could be simplified in sections like the discussion. Some scientific terminology is overly technical for a general reader, so clarification may be needed in specific areas.

Spelling and grammatical issues: There are a few minor errors, such as "re suspended" (should be "resuspended"), "sacarose" (should be "sucrose"), and "permaebilization" (should be "permeabilization").

5.1. We appreciate your detailed feedback on the article and suggestions. The discussion has been carefully revised to reflect the suggested changes, including simplifying sentences and scientific terms to make them understandable to general readers. In addition, we have corrected the spelling and grammar errors mentioned.

6. Ethical Standards and Research Integrity:

The study adheres to ethical standards, although there is no explicit mention of ethics approval, which should be included if necessary for any animal or human-related procedures (although this may not apply here).

There are no competing interests or financial disclosures that affect the research.

6.1. In the case of this study, ethical standards have been followed, but ethical approval does not apply as it does not involve procedures with animals or human beings. We have also confirmed that there are no conflicts of interest or financial disclosures related to the research.

7. Data Availability and Reporting Standards:

The authors have confirmed that all data necessary to replicate the study are included, and appropriate figures and tables support the data.

7.1. Yes, all the data necessary to replicate the study are duly included in the manuscript. The figures and tables have been carefully prepared to support and illustrate the data presented, ensuring the transparency and reproducibility of the results.

* Recommendations for Improvement:

- Clarification in Methods: Improve the explanation of the electroporation parameters, including the exact number of pulses and pulse duration.

- Language Refinement: Fix minor grammatical and structural issues to enhance clarity.

- Further Interpretation: Expand on the implications of the findings in a clinical context, and discuss potential limitations or challenges in applying electroporation for AK treatment.

- Ethics Approval: Ensure that any necessary ethical statements are included, if applicable.

*All the points mentioned were addressed. The explanation of the electroporation parameters is now improved, including greater detail on the number of pulses and their duration. We have also corrected grammatical and structural problems to ensure greater clarity in the text.

In addition, we expanded the interpretation of the results, discussing their implications in the clinical context.

As for ethical approval, it is not required as the study was not carried out on humans or animals.

---

## [Decision Letter · Decision Letter 2]

30 Dec 2024

Effects of electroporation on Acanthamoeba

polyphaga

PONE-D-24-12061R2

Dear Dr. Palloma Santiago Prates Pessoa,

We’re pleased to inform you that your manuscript has been judged scientifically suitable for publication and will be formally accepted for publication once it meets all outstanding technical requirements.

Kind regards,

Alireza Badirzadeh

Academic Editor

PLOS ONE

Additional Editor Comments (optional):

Reviewers' comments:

Reviewer's Responses to Questions

**Comments to the Author**

1. If the authors have adequately addressed your comments raised in a previous round of review and you feel that this manuscript is now acceptable for publication, you may indicate that here to bypass the “Comments to the Author” section, enter your conflict of interest statement in the “Confidential to Editor” section, and submit your "Accept" recommendation.

Reviewer #2: All comments have been addressed

2. Is the manuscript technically sound, and do the data support the conclusions?

Reviewer #2: Yes

3. Has the statistical analysis been performed appropriately and rigorously?

Reviewer #2: Yes

4. Have the authors made all data underlying the findings in their manuscript fully available?

Reviewer #2: Yes

5. Is the manuscript presented in an intelligible fashion and written in standard English?

Reviewer #2: Yes

6. Review Comments to the Author

Reviewer #2: "I have no concerns regarding dual publication or research ethics. The manuscript appears to comply with all necessary ethical standards. I have reviewed the submission and confirm that there is no indication of dual publication or unethical practices. The study follows appropriate guidelines for ethical research, and no issues related to research ethics were noted."

7. PLOS authors have the option to publish the peer review history of their article (what does this mean? ). If published, this will include your full peer review and any attached files.

**Do you want your identity to be public for this peer review?** For information about this choice, including consent withdrawal, please see our Privacy Policy .

Reviewer #2: **Yes: ** Ahmad Hosseini-Safa

---

## [Editor Report · Acceptance letter]

PONE-D-24-12061R2

PLOS ONE

Dear Dr. Pessoa,

I'm pleased to inform you that your manuscript has been deemed suitable for publication in PLOS ONE. Congratulations! Your manuscript is now being handed over to our production team.

Kind regards,

on behalf of

Dr. Alireza Badirzadeh

Academic Editor

PLOS ONE